# MicroRNA Analysis of Human Stroke Brain Tissue Resected during Decompressive Craniectomy/Stroke-Ectomy Surgery

**DOI:** 10.3390/genes12121860

**Published:** 2021-11-23

**Authors:** Andrew P. Carlson, William McKay, Jeremy S. Edwards, Radha Swaminathan, Karen S. SantaCruz, Ron L. Mims, Howard Yonas, Tamara Roitbak

**Affiliations:** 1Department of Neurosurgery, University of New Mexico Health Science Center, Albuquerque, NM 87131, USA; AndrewCarlson@salud.unm.edu (A.P.C.); William.mckay@ucdenver.edu (W.M.); rmims@uw.edu (R.L.M.); Howardyonas1@gmail.com (H.Y.); 2Department of Chemistry and Chemical Biology, University of New Mexico, Albuquerque, NM 87131, USA; jsedward@unm.edu (J.S.E.); rswaminathan@unm.edu (R.S.); 3Department of Pathology, University of New Mexico Health Science Center, Albuquerque, NM 87131, USA; KSantaCruz@salud.unm.edu; 4Department of Neurology, University of New Mexico Health Science Center, Albuquerque, NM 87131, USA

**Keywords:** stroke, malignant hemispheric infarction, microRNA, next generation microRNA sequencing

## Abstract

Background: Signaling pathways mediated by microRNAs (miRNAs) have been identified as one of the mechanisms that regulate stroke progression and recovery. Recent investigations using stroke patient blood and cerebrospinal fluid (CSF) demonstrated disease-specific alterations in miRNA expression. In this study, for the first time, we investigated miRNA expression signatures in freshly removed human stroke brain tissue. Methods: Human brain samples were obtained during craniectomy and brain tissue resection in severe stroke patients with life-threatening brain swelling. The tissue samples were subjected to histopathological and immunofluorescence microscopy evaluation, next generation miRNA sequencing (NGS), and bioinformatic analysis. Results: miRNA NGS analysis detected 34 miRNAs with significantly aberrant expression in stroke tissue, as compared to non-stroke samples. Of these miRNAs, 19 were previously identified in stroke patient blood and CSF, while dysregulation of 15 miRNAs was newly detected in this study. miRNA direct target gene analysis and bioinformatics approach demonstrated a strong association of the identified miRNAs with stroke-related biological processes and signaling pathways. Conclusions: Dysregulated miRNAs detected in our study could be regarded as potential candidates for biomarkers and/or targets for therapeutic intervention. The results described herein further our understanding of the molecular basis of stroke and provide valuable information for the future functional studies in the experimental models of stroke.

## 1. Introduction

Ischemic stroke accounts for up to 71% of all stroke cases and 51% of all stroke-related deaths worldwide, and is currently among the top leading causes of serious, long-term disability [1,2]. Identification of the molecular markers of stroke provides the potential to predict severity and clinical outcome, as well as to identify possible targets for therapeutic intervention. Stroke-associated ischemic damage involves blood brain barrier (BBB) dysfunction, microvascular injury, cytotoxic and vasogenic edema, post-ischemic inflammation, oxidative damage, spreading depolarization, and ultimately, the death of neurons, glia, and endothelial cells [3,4,5,6,7]. In search of the molecular mechanisms and signaling cascades regulating these pathological processes in human subjects, researchers focus on the differential gene profiling in peripheral blood and CSF collected from stroke patients [8,9,10]. Besides the identified alterations in gene expression, stroke-associated changes in microRNA profiles are now regarded as indicators of the risk, occurrence, and severity of the disease [11,12,13]. 

MicroRNAs (miRNAs) are a diverse class of highly conserved small RNA molecules that function as critical regulators of gene expression and orchestrate a variety of signaling pathways involved in stroke progression [14,15]. Among the wide range of identified miRNAs, some are now considered as biomarkers for cerebral ischemia [13,14,16]. The stroke-related changes in human gene and miRNA expression were detected using the patient blood (whole blood and plasma), cerebrospinal fluid, or postmortem brain samples [17,18,19,20]. Even though these samples represent simple and clinically relevant alternatives to brain tissue, the interpretation of the obtained gene and miRNA expression profiling results remains problematic. While these data may predict a clinical outcome or describe the systems response to disease, they do not distinguish the brain tissue-specific alterations from the molecular changes associated with systemic immune response or postmortem transformations. 

In the present study, we analyzed human brain tissue samples resected during a craniectomy and stroke-ectomy procedure in patients with severe (malignant) hemispheric stroke. In up to 10% of stroke patients, the infarct and associated edema progresses into severe space-occupying brain swelling associated with a high mortality rate of up to 80% [21,22,23]. Malignant cerebral infarction caused by middle cerebral artery (MCA) or internal carotid artery (ICA) occlusion leads to an increased intracranial pressure and subsequent herniation and brain stem compression. The affected MCA territory includes a portion of the frontal and temporal/parietal lobes. Depending on the location of the occlusion, ICA-affected brain area could be smaller or wider than that of an MCA territory infarction. Recommended treatments for malignant edema include early identification of neurological worsening, osmotic therapy, and decompressive craniectomy (DC) with dural expansion in select patients [24,25,26,27]. Additional removal of infarcted tissue (stroke-ectomy) is performed in cases of severe swelling in order to quickly decompress deep structures [28,29]. This usually aspirated and discarded tissue provided a unique source for our human stroke tissue analysis. Herein, for the first time, morphological evaluation, as well as gene and microRNA profiling was performed in the freshly removed human stroke brain tissue. This approach enables the identification of early (as opposed to post-mortem) brain tissue-specific alterations in the live brain following cerebral stroke. The obtained data could contribute to knowledge on the underlying mechanisms of stroke, as well as lead to development of miRNA-based targeting therapies in the future. 

## 2. Materials and Methods

### 2.1. Tissue Sample Collection

The sample collection procedures and the roles of all participants in the study are regulated by the protocol approved by the University of New Mexico Human Research Review Committee (IRB approval: UNM-HRPO 17-031).

Surgical procedure: Brain tissue specimens were collected from the ischemic stroke patients with large cerebral infarction who underwent decompressive hemicraniectomy with stroke-ectomy procedure. During the surgery, a large (≥12 cm) craniectomy was created and the dura opened in a stellate fashion. After the dura was opened, the brain was inspected and if it is still herniating out of the defect, additional infarcted temporal lobe tissue was removed and stored immediately in RNAlater solution to stabilize the samples for further analyses. Additional stroke-ectomy was performed with suction and bipolar cautery until the brain was adequately relaxed. 

Sample handling: The resected tissue was taken to the UNM Human Tissue Repository and Tissue Analysis Shared Resource (HTR-TASR, an honest broker). Control temporal lobe samples of non-stroke patients were received from the HTR-TASR as frozen blocks. These biopsy samples were collected during a neuro-oncology surgery, either from the tissue adjacent to the tumor, or tissue removed when accessing a deep tumor. These temporal lobe samples were identified by the pathologist as “normal”, based on the absence of gliosis, necrosis, or tumor cells. Similar histologically normal tissue has been widely used as reference normal samples for genomic and proteomic analyses in cancer research [30,31].

### 2.2. Human Brain Tissue Processing

Before sectioning, the tissue specimens were cut longitudinally to obtain two mirror samples from the same tissue block. One portion of the brain tissue was fixed in 10% neutral buffered formalin and subjected to sectioning and histological staining, followed by morphological evaluation. Another “mirror” part of each sample was stored in RNA later and subsequently frozen at −70 °C, for further microRNA analysis.

### 2.3. Histological Staining 

4 μm paraffin-embedded tissue sections were cut using RM2135 microtome, Leica (Buffalo Grove, IL, USA). After the deparaffinization, the sections were subjected to hematoxylin and eosin (HE) staining. The staining was performed using a protocol for standard automated HE histopathology staining with Leica Autostainer XL.

### 2.4. Immunohistochemistry and Fluorescence Microscopy

Immunofluorescence staining was performed based on the modified protocols for human brain tissue labeling described earlier [32,33]. After the regular deparaffinization and rehydration steps and prior to immunohistochemical labeling, the human tissue samples underwent antigen retrieval by incubating sections for 10 min in a solution of Tris-buffered saline (TBS) containing 20 μg/mL proteinase K. For blocking and permeabilization, the sections were incubated for 2 h at room temperature in TBS with 0.05% tween (TBST) containing 2% normal goat serum. Incubation with primary antibody was performed for 24 h at room temperature. The following antibodies were used for the immunofluorescence labeling: mouse monoclonal pan-neuronal marker antibody (1:100, MilliporeSigma, Burlington, MA, USA), rabbit polyclonal anti-Iba-1 (1:500, Fujifilm, Wako Chemicals, Richmond, VA, USA), and mouse monoclonal anti-GFAP (1:100, BD Biosciences, Franklin Lakes, NJ, USA). FITC-, and Rhodamine-conjugated secondary antibodies (1:500 concentration, 2 h at room temperature incubation) were from Jackson Immunoresearch. DAPI staining was used to visualize nuclei. Imaging was performed using Zeiss LSM 800 Airyscan confocal microscope, using single-scan and tile-scan image acquisitions.

### 2.5. Next Generation Sequencing (NGS) Analysis

Tissue samples were sent to Qiagen (Carlsbad, CA, USA) for RNA isolation and miRNAseq library preparation. miRNA and small RNA sequencing was performed using Illumina NGS sequencing platform. The resulting NGS data were analyzed using the CLC Genomics Workbench (version 20.0.2) and CLC Genomics Server (version 20.0.2), developed by QIAGEN, Aarhus, Denmark. First, quality and adapter/common sequence trimming on the reads was performed, and the trimmed reads were grouped according to Unique Molecular Identifiers (UMIs) and aligned to miRbase v22. Reads were normalized for expression analysis using trimmed mean of M-values method (TMM). miRNA differential expression analysis was performed using the EdgeR Bioconductor package 3.14.

Whole Transcriptome RNA-Sequencing Analysis: The unmapped reads from the NGS miRNA analysis were extracted, deduplicated and mapped to the genome. Gene expressions were calculated by counting number of reads mapping to the annotated gene loci.

All NGS sequencing data have been deposited in NCBI’s Gene Expression Omnibus and are accessible through GEO Series accession number GSE155257 at: https://www.ncbi.nlm.nih.gov/geo/query/acc.cgi?acc=GSE155257 (accessed on 29 July 2020).

### 2.6. Statistical Analysis

Statistical analysis was performed using the EdgeR Bioconductor package (version 3.36). *p*-values for significantly differentially expressed miRNAs and genes were estimated by an exact test assuming a negative binomial distribution. As a result, a list of miRNAs and their target genes with differential expression between the stroke and control tissue samples were generated and further analyzed to interpret their biological significance. The Empirical analysis of gene expression data was implemented as the ‘Exact Test’ for two group comparisons incorporated in the EdgeR Bioconductor package. This test was implemented to calculate miRNA differential expression, in particular, for calculations of “Log-Fold-Change” and “FDR corrected *p*-values” which are the main result of the statistical testing. Only the miRNAs and genes with 2-fold change in expression in Stroke group (compared to Control group) and FDR *p* < 0.05 are reported as differentially expressed. The volcano and PCA plots were visualized using the Enhanced Volcano Bioconductor package (version 1.12.0) and EdgeR Bioconductor package respectively.

### 2.7. Evaluation of miR-155 Target Genes 

Human miR-155 Targets RT^2^ Profiler PCR Array (Qiagen) was used to assess the expression of currently known experimentally verified and bioinformatically predicted 84 human genes regulated by miR-155. Three RNA samples per stroke and control groups were evaluated. The PCR Array Data analysis was performed using an automated PCR Analysis Web Portal and GeneGlobe Data Analysis (Qiagen). The *p*-values are calculated using a Student’s *t*-test followed by a Benjamini-Hochberg correction method.

### 2.8. Bioinformatics Analysis of miRNA Sequencing, Whole Transcriptome RNA-Sequencing, and PCR Array Data 

For each differentially expressed miRNA, the experimentally validated and predicted target gene list was generated using a multimiR package. Subsequently, a degree of association between miRNA target genes and specific GO terms from GO Biological Process domain was determined using the Gene Ontology Consortium Resource [34,35,36]. GO analyses were applied to determine the functional meaning of the miRNA sequencing results by detecting: (1) comprehensive sets of functional annotation tools to identify enriched biological processes, (2) relations that operate between the identified GO terms, and (3) association of miRNAs and genes with specific disease. To investigate the signaling pathways mediated by the target genes for each of the differentially expressed miRNA, GO using “Biological Process” and “Molecular Function” were performed. Only pathways with an FDR-adjusted *p* < 0.05 are represented. Additional KEGG pathway enrichment analysis from Database for Annotation, Visualization and Integrated Discovery (DAVID) bioinformatics tool was applied to determine molecular pathways and biological processes associated with miR-155 target genes.

## 3. Results

Brain tissue specimens from five non-consecutive patients with malignant hemispheric stroke and three non-stroke patients were used in our study. Patients were selected based on the need for stroke resection for severe swelling in the opinion of the attending neurosurgeon. Detailed information about the stroke and control samples is provided in Appendix A. Representative neuroimaging is shown in Figure 1. The figure describes a typical case with pre-operative and post-operative images demonstrating the large region of stroke (Figure 1A,B) and subsequent stroke tissue removal (Figure 1C).

### 3.1. Morphology of the Brain Tissue Collected from Stroke Patients

#### 3.1.1. Brain Tissue Damage

The representative images of the temporal lobe tissue obtained from non-stroke control and stroke patient (HE staining) are demonstrated in Figure 2 and Appendix A. The tissue from non-stroke patient cortex did not present any signs of edema or vascular damage, and comprised normal neurons with abundant lightly eosinophilic cytoplasm and good nuclear detail with prominent nucleoli (Figure 2A,B). All stroke patient samples were characterized by classic hypoxic-ischemic changes including bright eosinophilic cytoplasm and lack of the nuclear detail in neurons. The tissue was characterized by acute neuronal degeneration and classic appearance of the “eosinophilic neurons” (also termed “red” neurons), identified by cell body shrinkage, darkly stained pyknotic nuclei, and an intensely stained red eosinophilic cytoplasm (black arrows, Figure 2C,D). While all pyramidal neurons were severely damaged, some of them had a relatively intact chromatin pattern, and others comprised the necrotic “ghost” pyramidal neurons with absent affinity for hematoxylin and nuclei (blue arrows, Figure 2D). Peri-vascular and peri-neuronal widening was consistent with edema (stars, Figure 2D).

#### 3.1.2. Leukocyte Infiltration

Pathological processes associated with stroke are exacerbated following hypoxia-induced blood-brain barrier damage and infiltration of different types of leukocytes, including macrophages and neutrophils. Recruitment of circulating leukocytes in the ischemic brain contributes to neuroinflammation, BBB damage, and further loss of brain tissue following stroke [4]. Neutrophils are among the first cells infiltrating the brain after ischemic stroke; they are detected in the blood vessels within first 7 h and peak at 1 to 3 days after stroke offset. Monocyte infiltration is detected within the first 24 h post-ischemia, peak at 4 days, and some of these cells persist for weeks and acquire features of tissue macrophages [37,38]. We did not observe any notable monocyte/macrophage infiltration in the obtained samples, which was expected based on the early (17–72 h after stroke onset) time point of tissue collection. We, however, detected vascular damage and so called neutrophil margination and “pavementing” along the endothelium, a process that occurs before the leukocyte extravasation (Figure 2E,F). A significant accumulation of neutrophils, both in the perivascular space and in the vicinity of the blood vessels was observed (arrowheads, Figure 2E,F). Neutrophils were also detected in the brain parenchyma, at a considerable distance from the blood vessels (arrowheads, Figure 2G,H). This observation is important given the controversy around the question of whether neutrophils are detectable in the stroke brain parenchyma [38,39].

**Figure 2 genes-12-01860-f002:**
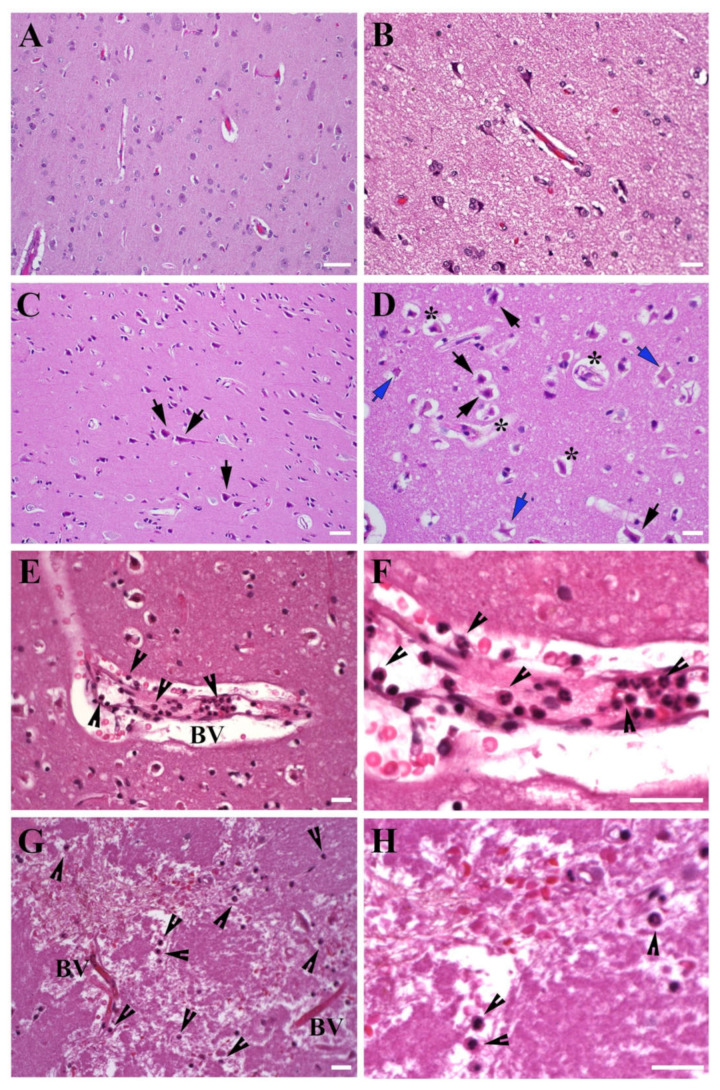
Morphology of human stroke brain tissue. Morphological evaluation of the temporal lobe tissue, HE staining. (**A**,**B**), Control non-stroke samples; (**C**,**D**), Brain tissue collected at 48 h after stroke onset. Black arrows: eosinophilic (red) neurons with darkly stained pyknotic nuclei. Blue arrows: necrotic “ghost” neurons with an absent nuclear detail. Stars: peri-neuronal and peri-vascular space widening associated with edema. Neutrophils (arrowheads) were detected in the vicinity of large vessels and capillaries (**E**,**F**), tissue collected at 60 h after stroke onset), as well as in the brain parenchyma (**G**,**H**), tissue collected at 28 h after stroke onset). (**E**) and higher magnification in (**F**): Margination and pavementing of neutrophils along the blood vessel wall; some neutrophils migrated out and are detected in the blood vessel vicinity. (**G**) and higher magnification in (**H**): Neutrophils are detected at a considerable distance from the blood vessels, which demonstrates the process of neutrophil invasion into the damaged brain tissue. BV- blood vessel. Bars: (**A**,**C**): 50 μm; (**B**,**D**,**E**–**H**): 20 μm.

#### 3.1.3. Immunofluorescence Microscopy 

Immunofluorescence staining of the same stroke tissue samples demonstrated that small (unipolar and bipolar) cortical neurons (Figure 3A) and spindle-like neurons (Figure 3B) retained their morphology and cytoarchitecture. Pan-neuronal marker antibody staining revealed a sophisticated neuronal network with still intact processes and intercellular connections (Appendix A). Astrocytes were characterized by the low GFAP expression and damaged appearance (Figure 3C), while microglia were represented by normal looking spindle-shaped cells (Figure 3D).

To summarize, morphological evaluation demonstrated that human stroke samples used in our study represented a severely damaged cortical tissue characterized by early ischemic changes. While tissue edema was intense and tissue damage was irreversible, the observed morphological and cytological characteristics pointed to a transitional stage between a severe acute hypoxic injury and tissue necrosis.

### 3.2. miRNAs Differentially Expressed in Stroke Brain Tissue

NGS analysis of stroke and non-stroke control human brain tissue specimens was performed to assess stroke-related changes in expression of 1815 miRNAs. At the first stage of elimination from the list of miRNAs dysregulated in stroke tissue, a *p* < 0.05 cutoff resulted in a group of 172 miRNAs with impaired (>1.5 fold increased or decreased) expression in stroke patient vs. non-stroke control samples. Of these miRNAs, 75 (58 upregulated and 17 downregulated) miRNAs were previously reported to be differentially expressed in the whole blood, serum, or CSF samples from stroke patients (Appendix A [40,41,42,43,44,45,46,47,48,49,50]). Further FDR adjusted *p*-value cutoff resulted in a list of 34 miRNAs differentially expressed in stroke brain tissue. Volcano plot on Figure 4A displays magnitude changes of miRNAs in stroke vs. control samples. Principal Component analysis (PCA) was used to reduce the dimension of large data sets and explore sample clusters arising naturally based on the expression profile (Figure 4B).

PCA analysis demonstrated that control samples were segregated into a separate cluster, while, with the exception of sample 4, stroke samples clustered together. A portion of a heat map on Figure 4, C shows the profiles of 50 miRNAs, demonstrating a similarity between the control samples from three different non-stroke patients. miRNA expressions were not statistically different between control samples.

Of the detected 34 differentially expressed miRNAs, 19 (17 upregulated and 2 downregulated) were identified as stroke-related miRNAs with >2-fold changed expression in stroke samples. Based on the available literature, the aberrant expression of these miRNAs and their family members is associated with stroke progression and outcome (Table 1). miRNA-based regulation of various cellular and biological processes is mediated via their direct target genes and proteins. To explore the functional significance of the differentially expressed miRNAs, we identified their experimentally verified and predicted target genes and subsequently applied GO pathway enrichment analyses, as described in Methods. In agreement with previous reports, our analyses detected that biological processes significantly influenced by the identified miRNAs (including neuronal death, inflammatory response, blood coagulation, glutamate and acetylcholine secretion, synaptic plasticity, and vascular permeability) are critical for stroke progression and outcome. Importantly, the upregulation of miR-1246, miR-4516, miR-320a-3p, miR-320c, miR-204-3p, miR-17-5p, miR-16-5p, and miR-423-5p detected in stroke brain tissue is in agreement with the reported induction of these miRNAs and their family members in stroke patient blood or CSF [51,52,53,54,55,56,57,58]. 

#### Newly Detected Group of miRNAs Dysregulated in Stroke Brain Tissue

NGS analysis of stroke and non-stroke patient samples detected 15 differentially expressed miRNAs with no previously reported association with human stroke (Table 2). The list included 5’ or 3’ strands of the mature miRNAs.

Bioinformatic analysis using the GO database revealed the association of this group of miRNAs with multiple diseases and pathological conditions, including coronary artery disease, stroke, amyotrophic lateral sclerosis, and Parkinson’s disease. Functional GO pathway enrichment analyses detected a significant association of these miRNAs with various stroke-related biological processes. As summarized in Figure 5, the identified miRNAs are significantly associated with biological processes and signaling pathways, which play an important role in stroke progression and recovery.

Appendix A demonstrates some of the multiple predicted and experimentally validated target genes regulated by the identified miRNAs. In order to validate the dysregulation of newly detected miRNAs, we assessed the expression of their known direct target genes in stroke and control brain tissue. Table 3 presents the list of the miRNAs and their target genes; 20 genes with the greatest up- or downregulation in stroke vs. control tissue are displayed. As expected, microRNAs and their target genes had the opposite expression pattern; a number of genes were targeted by several newly detected microRNAs (Table 3, stars). We were unable to verify the expression of target genes for miR-10395-3p, miR-9901, and miR-412, due to a very limited number of their known predicted targets. Among the deregulated target genes were: *SMAD* genes implicated in regulating inflammation, myelination, and vascular function after stroke [62]; *WNT1*, the regulator of Wnt signaling pathway mediating neurogenesis and vascular remodeling [63]; *MAPK81P2*, a critical activator of mitogen-activated protein kinase (MAPK) pathway contributing to neuroinflammation and neuronal survival after stroke [64]; *NOTCH2* influencing neuronal viability [65]; *PDGFRA*, regulating platelet-derived growth factor signaling contributing to post-stroke atherosclerosis [66]; *ADAMTS5*, a member of matrix metalloproteinase family, regulating degradation of extracellular matrix proteins, tissue repair, and remodeling [67]; *FGF12,* a member of fibroblast growth factor (FGF) family implicated in neuroprotective effect after stroke [68]; and *NOS1* encoding neuronal nitric oxide synthase 1, associated with stroke-induced neurotoxicity [69].

To summarize, based on the performed bioinformatic and RNA-sequencing analyses, we concluded that the aberrant expression of the newly detected miRNAs has a significant functional implication and thus, these miRNAs could prominently influence the progression and outcome of human stroke.

### 3.3. miR-155 Expression in Stroke Brain Tissue

Due to our long-standing interest in the role of miR-155 in human stroke, we performed a separate investigation of its deregulation in stroke brain tissue. miR-155 is an evolutionarily conserved multifunctional miRNA, expressed in all human tissues and implicated in contributing to various pathological processes [70,71]. A number of our previous studies have been focused on the role of this miRNA in ischemic stroke. In a rodent model of stroke, an elevated miR-155 expression in the brain was linked to higher stroke severity, while miR-155 inhibition reduced the infarct size, preserved BBB integrity, and supported the functional recovery [72,73]. In the present study, miR-155 was significantly (>1.5-fold, *p* = 0.04) elevated in human stroke tissue (Appendix A). However, due to a high variability among five stroke samples, an average miR-155 transcript expression change did not reach the significance cutoff after the FDR *p* < 0.05 adjustment. Interestingly, miR-155 levels tended to be higher in samples S1, S2 and S5 collected within 48–72 h after stroke onset, and lower in samples S3 and S4 collected within 17–28 h after stroke. In order to evaluate stroke-related changes of miR-155 activity, we performed miR-155 target gene profiling of samples S1, S2 and S3 (Figure 6). miR-155 Targets PCR Array analysis detected a significant (>1.5-fold, FDR *p* < 0.05) downregulation of 50 miR-155 direct target genes in stroke samples compared to controls (Figure 6A,C, blue). Interestingly, a significant decrease in target gene expression was also detected in the S3 patient sample characterized by a moderate increase in miR-155 (Figure 6B). Based on the whole transcriptome data, additional 23 miR-155 target genes (not included in the PCR Array panel) were significantly downregulated in stroke samples (Figure 6A, green). GO and KEGG pathway enrichment analyses revealed that the detected group of genes is involved in the regulation of critical signaling pathways, including TGF-β, ERK, NF-kB, Wnt, MAPK, ErbB, Ras, PI3K-Akt, and VEGF signaling cascades. Moreover, bioinformatic analysis identified several stroke-related and brain-specific biological processes regulated by this set of genes, including glial cell differentiation, endothelial morphogenesis, leukocyte activation, macrophage cytokine production, neurodegeneration, and synaptic function. Raw data demonstrating miR-155 direct target expression are uploaded in GEO database. Based on the obtained results, we concluded that: (1) even moderate miR-155 expression changes induce a significant downregulation of its direct targets, and (2) 73 detected genes differentially expressed in stroke tissue could be regulated on the RNA level via miR-155-induced mRNA degradation.

## 4. Discussion

The underlying mechanisms of malignant cerebral infarction include general cellular and molecular processes associated with all types of strokes. Besides the known stroke-induced pathophysiological cascades leading to brain tissue damage and BBB disruption, additional contributing factors to malignant infarction and brain swelling include the involvement of the autonomic nervous system [24,74], and cortical spreading depolarization [22,75,76,77]. Based on the morphological evaluation of freshly removed stroke tissue, we propose that miRNA sequencing results obtained in our study could reflect molecular mechanisms attributed to (1) stroke-associated brain edema and tissue damage, and (2) transition from ischemic damage and selective neuronal death to tissue necrosis and infarction. We propose that our findings contribute to better understanding of these concomitant processes occurring at the early stages of stroke.

The main limitation of our study is the small number of analyzed samples. The reason for this is that DC with stroke-ectomy is a very rare procedure that is restricted to a select group of patients. To minimize the data variability, we analyzed only the temporal lobe samples from patients with an MCA/ICA territory infarction. This further reduced the number of specimens included in the analysis. To control false discovery rate, greater than 2-fold change and FDR *p* < 0.05 cutoff were applied to all miRNA and mRNA sequencing data reported in Table 1, Table 2, and Table 3. We believe that low FDR *p*-values, which often ranged between 1.61 × 10^−10^ and 0.01, are the indicators of the strength and reliability of our study.

### 4.1. Dysregulation of miRNAs with Known Association with Stroke

Our investigation established a strikingly similar expression pattern of 8 miRNAs detected in our brain tissue samples and blood/CSF samples reported by other groups. Therefore, we propose that the impaired expression of these miRNAs is associated not only with malignant infarction, but with cerebral stroke in general. Based on our analysis and reports by other groups, miR-1246, miR-4516, miR-320a-3p and miR-320c, miR-204-3p, miR-17-5p, miR-16-5p, and miR-423-5p, were upregulated both in stroke brain tissue and patient blood/CSF. Thus, signatures of these miRNAs in the blood or CSF samples accurately reflect their dynamics in the brain tissue. We, therefore, propose that these miRNAs could be regarded as potential candidate biomarkers for stroke prognosis and outcome. This statement is supported by the data from previous studies demonstrating the functional significance of several miRNAs from the identified group. miR-1246 is dramatically increased in CSF of patients with larger infarction [53]. Upregulation of miR-320 family miRNAs is associated with the modulation of aquaporin family proteins and stroke-induced edema [55,78,79]. Increased plasma miR-16 is associated with large infarction [58]. Based on these reports and our data, we propose that profiling of these microRNAs in the blood or CSF could provide a timely and valuable information for choosing a potentially successful therapeutic approach. 

Among the identified significantly dysregulated miRNAs, miR-182-5p, miR-183-5p, miR-320b and d, miR-505-5p, and miR-196a-5p had an opposite shift in expression levels in our stroke brain tissue samples and blood samples detected by other groups. These discrepancies reflect the dynamics of miRNA exchange between the brain tissue and circulating blood, and could be associated with a different sample collection time. This interesting observation will be addressed in our future studies where miRNA profiling will be performed in the brain tissue and blood samples from the same patient. Due to a long-standing interest in miR-155, we performed a separate investigation of its activity in human stroke samples. Our findings revealed that, despite the moderate increase in miR-155 expression in some samples, activity of this miRNA resulted in a significant downregulation of 72 direct target genes. Based on bioinformatics analysis, 30 of the detected genes represent the shared targets between miR-155 and at least one miRNA from the list provided on Table 1. Therefore, further investigation is needed to properly evaluate miR-155 activity in the stroke brain tissue. Dysregulation of miR-155 and its possible correlation with the time of stroke onset will be a subject of our upcoming study involving a larger group of stroke patients.

### 4.2. Newly Detected miRNAs Dysregulated in Human Stroke Tissue

A group of miRNAs differentially expressed in stroke brain tissue included 15 significantly dysregulated miRNAs with no previously reported changes in stroke patient blood or CSF. Most of them are the recently identified miRNAs with yet unexplored functions. Based on our bioinformatics approach, they can regulate molecular processes that are critical for stroke progression. We suggest that dysregulation of these newly detected miRNAs could reflect: (1) brain tissue-specific changes, and (2) miRNA signatures specific to malignant infarction and brain swelling. The role of this novel group of miRNAs will be further investigated in the functional studies utilizing in vitro or animal models of stroke.

## 5. Conclusions

In this first analysis of human stroke brain tissue biopsy samples, we described morphology and cytoarchitecture of the temporal lobe cortex at 17–72 h after the hemispheric infarction. microRNA profiling detected 34 miRNAs aberrantly expressed in stroke-tissue, which could influence stroke progression and outcome. We anticipate that the obtained data will contribute to knowledge of the molecular basis of stroke, and provide a foundation for the miRNA-based intervention strategies focused on stroke outcome and prevention of malignant brain swelling.

## Figures and Tables

**Figure 1 genes-12-01860-f001:**
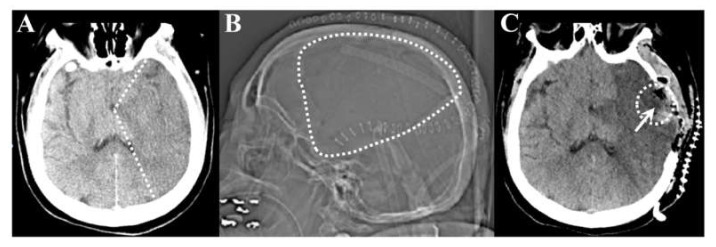
Representative pre-operative and post-operative images from the same stroke patient. (**A**): Axial computed tomography (CT) scan at one day post stroke demonstrating left sided middle cerebral artery infarct with loss of grey-white matter differentiation in a large territory. The white dotted line points out the approximate infarct border. (**B**): Post-operative CT scout image demonstrating the large craniotomy defect (dotted line). (**C**): Post-operative axial CT scan demonstrating completed infarct, bony decompression, and the region of partial stroke resection in the temporal lobe (arrow).

**Figure 3 genes-12-01860-f003:**
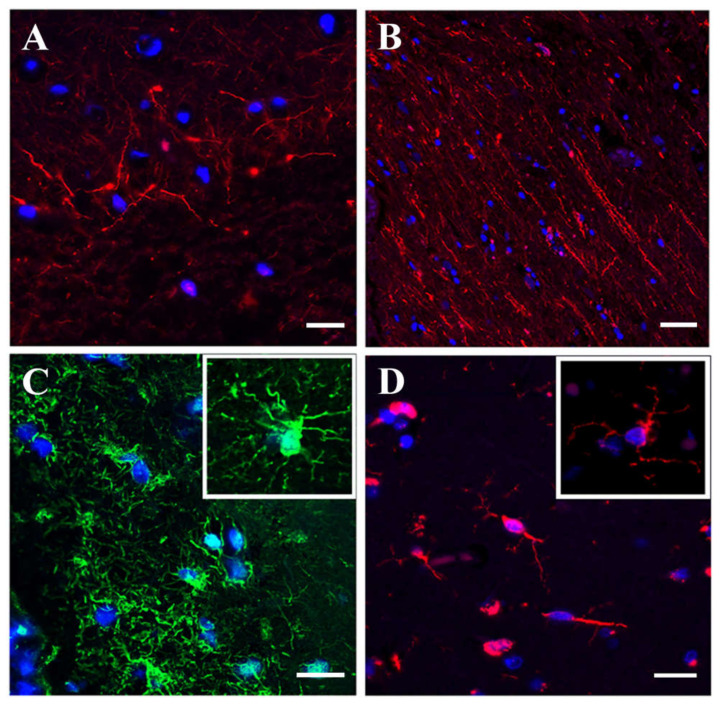
Immunofluorescence analysis of human stroke brain tissue IF staining of human stroke brain tissue collected at 48 h after stroke onset. A, B: Pan-neuronal antibody staining (red) visualizing small unipolar and bipolar (**A**) and spindle-shaped (**B**) neurons. (**C**): GFAP (green) immunostaining for astrocytes. (**D**): Iba-1 (red) immunostaining for microglia. Bars: (**A**,**C**,**D**): 20 μm, and (**B**): 50 μm.

**Figure 4 genes-12-01860-f004:**
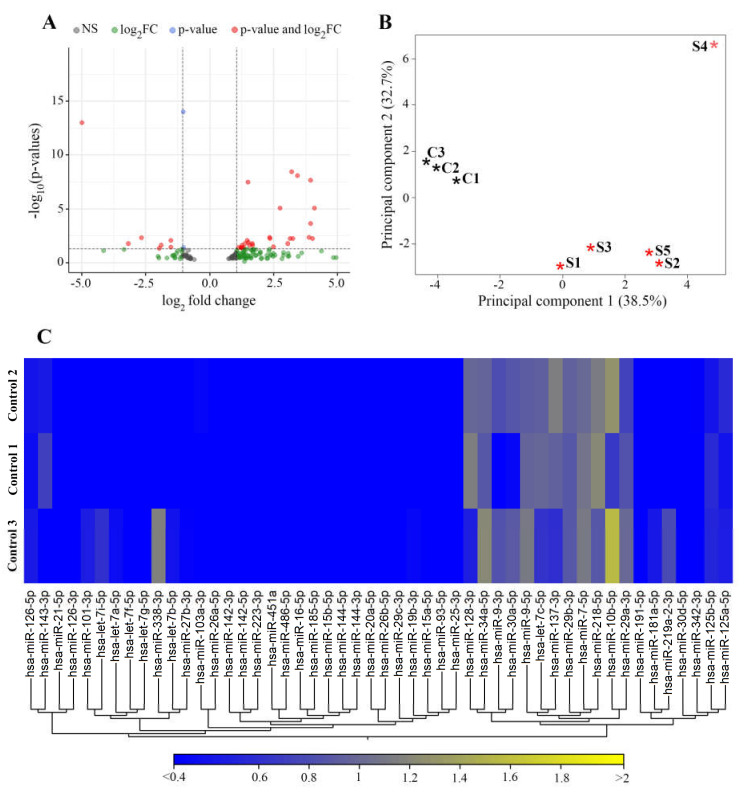
Visualization of miRNA expression in stroke and control tissue. (**A**): Volcano plot showing the magnitude of the difference in expression values of the samples in Control and Stroke groups. The graph is constructed by plotting the −log_10_(FDR corrected *p*-value) on the y-axis, and the log_2_(fold-change) on the x-axis from the group of 172 miRNAs with impaired expression in stroke patient vs. control samples. Grey circles represent samples with FDR *p*-value > 0.05, green circles represent miRNAs that have a fold-change > 1.5 and FDR *p*-value > 0.05, blue circles represent miRNAs that have an FRD *p*-value < 0.05 and a fold-change > 1.5, and red circles represent miRNAs that have an FDR *p*-value < 0.05 and fold-change >2.0. (**B**): Principal component analysis (PCA) plot for stroke and control samples. The PCA was performed on 34 miRNAs identified to be differentially expressed between stroke and non-stroke control samples using the miRNAs that have the largest coefficient of variation based on TMM normalized counts. Each asterisk represents a sample. Red asterisk–stroke samples from patients S1–S5; black–control non-stroke samples C1–C3. (**C**): A heat map demonstrating expression levels of 50 microRNAs in three different control samples. Each column represents one miRNA and each row represents one sample. The color represents the difference of the count value to the row mean. N = 5 stroke and 3 control samples.

**Figure 5 genes-12-01860-f005:**
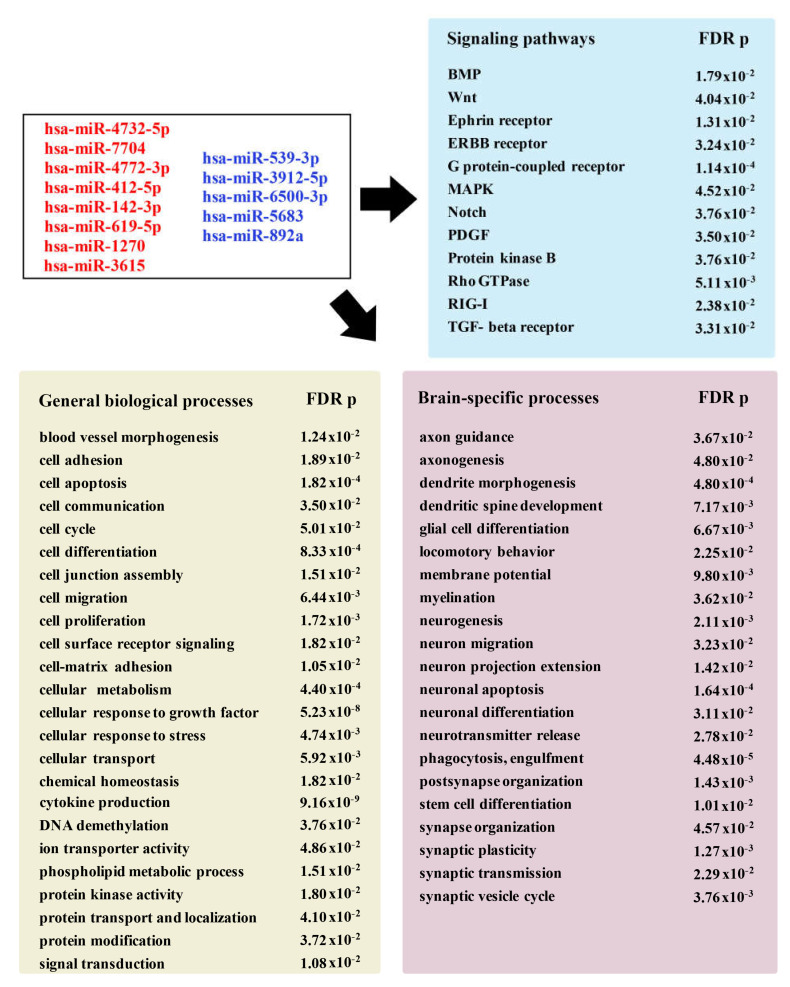
Pathway enrichment analysis for newly detected miRNAs. Diagram showing selected groups of GO terms (with the corresponding FDR *p*-value) significantly associated with the identified miRNAs (depicted in the box). The annotations describe molecular functions and signaling pathways, as well as general and brain-specific biological processes. Abbreviations: BMP- bone morphogenetic protein; MAPK—mitogen-activated protein kinase; PDGF—platelet-derived growth factor; RIG-I—retinoic acid-inducible gene receptor I; TGF-beta—transforming growth factor beta.

**Figure 6 genes-12-01860-f006:**
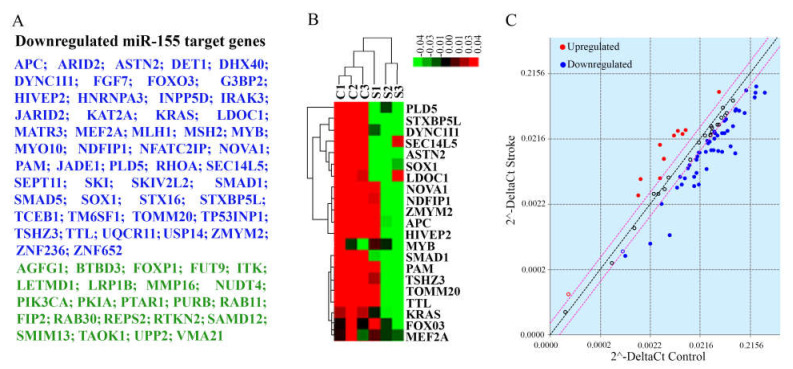
miR-155 target gene analysis. (**A**) A list of miR-155 direct target genes significantly downregulated in stroke tissue, compared to control samples. Blue—genes identified by PCR Array, green—additional genes identified by whole transcriptome sequencing. (**B**) A portion of a heatmap (after non-hierarchical clustering) displays a group of target genes with significantly lower (green) expression in the stroke samples. The heat map was generated based on delta Ct values. Genes with higher expression levels are shown in red, whereas genes with lower expression levels are shown in green. Genes with average expression levels are shown in black. (**C**) Scatter plot compares the normalized gene expression between the stroke and control samples. The central line indicates unchanged gene expression. Red circles identify genes significantly upregulated in stroke tissue, blue—genes significantly downregulated in stroke group. The graph plots the log10 of normalized gene expression levels in control samples (x-axis) versus stroke samples (y-axis). N = 3 samples per control and stroke groups.

**Table 1 genes-12-01860-t001:** Differential (>2-fold change, FDR *p* < 0.05) expression of miRNAs in human stroke brain tissue—significantly dysregulated stroke-related miRNAs. Red—upregulated and blue—downregulated miRNAs.

Name	Fold-Change	*p*-Value	FDR *p*-Value	Reported in:
hsa-miR-1246 hsa-miR-4516 hsa-miR-182-5p hsa-miR-320d hsa-miR-1255b-5p hsa-miR-320c hsa-miR-183-5p hsa-miR-196b-5p hsa-miR-204-3p hsa-miR-17-5p hsa-miR-193b-5p hsa-miR-16-5p hsa-miR-320b hsa-miR-423-5p hsa-miR-320a-3p hsa-miR-505-5p hsa-miR-652-3p hsa-miR-135a-3p hsa-miR-196a-5p	334.46 32.68 15.44 10.80 9.53 9.29 8.87 5.64 3.23 3.20 2.87 2.78 2.54 2.46 2.41 2.34 2.19 −9.00 −31.65	0 0 3.98 × 10^−6^ 5.73 × 10^−11^ 0.0001 2.30 × 10^−11^ 0.0001 0.0011 0.0008 0.0005 0.0004 0.0002 0.0017 0.0015 0.0008 0.0014 0.0015 0.000536 3.33 × 10^−16^	0 0 0.0002 8.67 × 10^−9^ 0.0059 3.79 × 10^−9^ 0.0059 0.0347 0.0251 0.0180 0.0142 0.0090 0.0465 0.0416 0.0263 0.0416 0.0416 0.0180 1.01 × 10^−13^	[51,52,53] [54] [18] [18,59] [60] [51] [18] [18] [57] [56] [61] [58] [18] [52] [55] [18] [18] [57] [18]

**Table 2 genes-12-01860-t002:** Differential (>2-fold change, FDR < 0.05) expression of newly detected miRNAs in human stroke brain tissue. Red—upregulated, blue—downregulated miRNAs.

Name	Fold-Change	*p*-Value	FDR *p*-Value
hsa-miR-10395-3p hsa-miR-4732-5p hsa-miR-7704 hsa-miR-9901 hsa-miR-4772-3p hsa-miR-412-5p hsa-miR-142-3p hsa-miR-619-5p hsa-miR-1270 hsa-miR-3615 hsa-miR-539-3p hsa-miR-3912-5p hsa-miR-6500-3p hsa-miR-5683 hsa-miR-892a	17.13 16.35 15.42 14.91 8.35 6.75 5.14 5.08 2.96 2.62 −2.08 −2.65 −2.83 −3.68 −3.88	1.00 × 10^−7^ 0.0002 1.61 × 10^−10^ 0.0001 0.0005 1.10 × 10^−7^ 0.0001 0.0001 0.0008 0.0005 0.0012 0.0001 0.0002 0.0007 0.0019	9.08 × 10^−6^ 0.0062 2.24 × 10^−8^ 0.0048 0.0172 9.08 × 10^−6^ 0.0060 0.0050 0.0254 0.0175 0.0363 0.0051 0.0090 0.0243 0.0514

**Table 3 genes-12-01860-t003:** Differentially (>2-fold change, FDR *p* < 0.05) expressed miRNA target genes in human stroke brain tissue. Red—upregulated and blue—downregulated miRNAs and genes. Target genes shared by at least 2 different miRNAs are marked with a star.

miRNA	Differentially Expressed Target Genes
hsa-miR-4732-5p	* UHMK1, TOX *, TMEM183A *, TMED8, TGOLN2, SMAD2, SAMD12, RORB, RAP2A, RALYL*, PURG *, PTCH1 *, PSD3 *, PRPF40A, PKIA *, PHKA1 *, PDE4D, PCDH17, PARM1, PAPOLA * *
hsa-miR-7704	* SPATA17, LDLRAP1, IWS1, DERL3, SEMA6A *, CACNA1E, ATG12, OPA1, PAK3, EPS8, CHMP3, POLR2L, GPR85, C4orf33, LETMD1, MEIS2 *, SPATA6, KLF12 *, ROBO1, PTCH1 * *
hsa-miR-4772-3p	* CTDSP2, TMEM183A *, PURA *, RHOU, CSRNP3, ARHGAP31 *, SH3BGRL2, CADM2 *, LRRC28 *, METTL15 *, PHKA1 *, TAOK1 *, GUCY1A2, MLLT3 *, CHIC1 *, RNF141 *, TOMM20 *, ZDHHC21 *, HS6ST3 *, PURG * *
hsa-miR-142-3p	* PDGFRA, TSPAN3, MTF2, FOXP1, REPS2 *, MTX3, MEIS2 *, PIK3CA, MAP2K6, CLSTN2, HMGCLL1 *, FGF12 *, SNTB2 *, NUCKS1, PURA *, CADM2 *, MAGI2, CHIC1 *, LRP1B, TOX * *
hsa-miR-619-5p	* EIF4EBP2, CADM2 *, LRRC28 *, METTL15 *, ATRNL1, NKAIN2, TAOK1 *, MLLT3 *, GALC, CHIC1 *, KDM3B, AMBRA1, IL1RAPL1, RNF141 *, TOMM20 *, RPS6KA6, ZDHHC21 *, KCNH5, HS6ST3 *, PURG * *
hsa-miR-1270	* LRRTM4, THADA, PCDH9, KLF12 *, STX17, MMP16, HMGCLL1 *, CNOT7, PURB, RALYL *, PTAR1, PKIA *, NOVA1, OLFM3, NUBPL, FGF12 *, SNTB2 *, PAPOLA *, DPP6 *, CNR1 *
hsa-miR-3615	* REPS2 *, SEMA6A *, ARHGAP15, MAMLD1, FOXB1, CCDC80, MEIS2 *, SLITRK5, PSD3 *, CBX5, DIRAS2, PKIA *, BTBD3, DPP6 *, ARHGAP31 *, TAOK1 * *
hsa-miR-539-3p	* AQP7 *, ADAMTS5, HAVCR1 *, GOLGA8G, ZNF266, TMCO5A, NOTCH2 *, PLCL1 *, HCN4, RYR2, PHKB *, ABCA13, LYN *, C18orf25, WIPF2, GATA3 *, FBLN2, TMPRSS4 *, TNFAIP3, ZC3H4 * *
hsa-miR-3912-5p	* FZD8 *, WNT1, PID1 *, ADAM8, GATA3 *, KLHDC7B, JAG2, DLST, RBMS3, FIG4, TMEM63A, IQSEC3 *
hsa-miR-6500-3p	* FZD8 *, HIC1, AQP7 *, MRGPRF, DMRT1, CTCFL, HSPA1A, UNC45B, SLCO5A1, SBSPON, FOXJ3, HAVCR1 *, PID1 *, NR5A1, COX20, NOXRED1, SCNN1G *, UBAP1, PLCL1 *, LYN * *
hsa-miR-5683	* LSM4, LOXL4, PPM1N, ARSI, GATA3 *, NUTM2D *, TMOD4, NLRP2, ZNF217, NGB, HOXD3, CYTH3, DACT3, CTPS2, MOB2, CD74, CLDN10, GALNT1, FZD8 *, MAPK8IP2 *
hsa-miR-892a	* NOTCH2 *, SCNN1G *, ACTG2, PHKB *, LYN *, ZBTB42, CASC4, NUTM2D *, FTCDNL1, IRX2, SLC16A3, RCL1, TMPRSS4 *, DPEP1, VSIG10, ZC3H4 *, SAP30BP, POTEG, NOS1, IRAK3 *

## Data Availability

All NGS sequencing and PCR array data have been deposited in NCBI’s Gene Expression Omnibus and are accessible through GEO Series accession number GSE155257 at: https://www.ncbi.nlm.nih.gov/geo/query/acc.cgi?acc=GSE155257 (accessed on 29 July 2021).

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
