# Peer review of "MicroRNA Analysis of Human Stroke Brain Tissue Resected during Decompressive Craniectomy/Stroke-Ectomy Surgery"

_genes, 2021, doi:10.3390/genes12121860_

Round 1

Reviewer 1 Report

Based on my scientific research experience I will provide my comments only on the data analysis and interpretation of bioinformatics analyses of the manuscript presented by Carlson et al

Carlson et al using brain samples taken from stroke patients and non-stroke patients detected transcriptional perturbance of hsa miRNAs.

Authors profiled miRNAs using outsource service based on Qiagen protocols for RNA isolation, sequencing library preparation and data analysis using the QIAseq miRNA quantification workflow.

Overall, the authors stated they have found 1,815 differentially expressed miRNAs between the two conditions. From the description provided in the paper, this group of miRNAs cannot be considered genuine DE because not filtered for FDR and fold-change.

In fact, authors did apply FDR correction and filtering and this lowered the potential number of DE miRNAs found (n=17). As results figure 4a is incorrect because does not show any filtering based on FDR and fold change. Also, the statistics related to the 1,815 DE miRNAs were no deposited as supplemental file.

Figure 4b shows the PCA of the dataset and clearly highlights the separation of the datapoints marking high variability in term of expression profiles among stroke patients. Based on this result and the low number of datapoints, I suggest to report only results derived by a sharp cutoff (FDR= 5% and fold change >=2) to exclude any false positive conclusion.

Table 1 reports the miRNAs passing some of above criteria n=19 miRNAs. Please refined this number by applying FDR and not pvalue cut-off.

To improve this results authors can use a batch-normalization protocol before calling for DE.

Figure4c reports the expression of miRNAs sequenced but does not show differences between groups.

GO and KEGG analyses are not described in the methods.

Authors extensively discussed the implication of has-miR155 in human stroke, but from this experiment miRNA 155 cannot be considered DE because not supported by statistical evidences: FDR= 0.4

Minor comments:

  • Line 295: figure 4c is flipped. Please correct in the manuscript the description: rows represent control condition while columns represent miRNAs.
  • Please define CSF
  • Please define which NGS platform was used to sequence the miRNAs
  • It is not clear which RNA molecules were sequenced. Only mature hsa miRNAs or the entire small RNAs fraction? Please specify in the method.
  • Also, it is not clear if authors performed whole transcriptome sequencing. GEO contains only data related to miRNAs and the repository it is not well documented.

Author Response

We thank the reviewer for careful and thoughtful evaluation of the manuscript. The point-by-point answers and concerns are addressed below:

Overall, the authors stated they have found 1,815 differentially expressed miRNAs between the two conditions. From the description provided in the paper, this group of miRNAs cannot be considered genuine DE because not filtered for FDR and fold-change. In fact, authors did apply FDR correction and filtering and this lowered the potential number of DE miRNAs found (n=17).

1) We apologize for the misleading information in the description, and we’ve made corrections to the text, where we now state that: “NGS analysis of stroke and non-stroke control human brain tissue specimens was performed to assess stroke-related changes in expression of 1,815 miRNAs”.

2) In the Methods section (statistical analysis), we’ve added the following: “Only the miRNAs and genes with 2-fold change in expression in stroke group (compared to control group) and FDR p<0.05 are reported as differentially expressed”.

3) FDR adjusted p-value cutoff resulted in a list of 19 (17 upregulated and 2 downregulated) stroke-related miRNAs with >2-fold changed expression in stroke samples.

4) We changed the manuscript text so that the term “Differential expression” applies only to the miRNAs and genes with >2-fold change and FDR p<0.05.

As results figure 4a is incorrect because does not show any filtering based on FDR and fold change. Figure 4b shows the PCA of the dataset and clearly highlights the separation of the datapoints marking high variability in term of expression profiles among stroke patients. Based on this result and the low number of datapoints, I suggest to report only results derived by a sharp cutoff (FDR= 5% and fold change >=2) to exclude any false positive conclusion.

The main purpose of  Figure 4 is to illustrate the magnitude of changes in miRNA expression between Stroke and Control groups. No conclusions on differential expression were drawn from the volcano and PCA plots, and all reports on DE of miRNAs are provided in the Tables.

Table 1 reports the miRNAs passing some of above criteria n=19 miRNAs. Please refined this number by applying FDR and not pvalue cut-off.

Only 19 miRNAs with >2-fold change, FDR p<0.05 are reported in Table 1

To improve this results authors can use a batch-normalization protocol before calling for DE.

Since all the samples were  processed and sequenced simultaneously and there were no technical differences in sample processing, we do not think that batch normalization is necessary in our case.  

Figure4c reports the expression of miRNAs sequenced but does not show differences between groups.

Figure 4C demonstrates that miRNA expression is similar in all control samples. No significant differences were detected. All the raw data are deposited online.

GO and KEGG analyses are not described in the methods.

Additional text was included to Methods section (Bioinformatic analysis)

Authors extensively discussed the implication of has-miR155 in human stroke, but from this experiment miRNA 155 cannot be considered DE because not supported by statistical evidences: FDR= 0.4

We fully agree with the reviewer and, accordingly, we made changes in the manuscript. Specifically, we removed miR-155 data from the section “3.2. miRNAs differentially expressed in stroke brain tissue”. We placed miR-155 data in a separate, last section of the manuscript, where we specifically emphasize moderate (not statistically significant) changes in the expression of this miRNA. Accordingly, we change the Figure numbers and order of appearance. For this reason, we made changes in the reference numbering in the text and reference list. We believe that miR-155 expression was not significant due to a small sample number. This is explained in Discussion section. The obtained data are very important for our research and would be of interest for other researchers. Therefore, we would like to retain these data in the present manuscript.

Reviewer 2 Report

Carlson et al. describe in the manuscript the morphological and miRNA expression changes in human brain samples obtained from severe stroke patients during craniectomy and brain tissue resection. Using histopathological and immunofluorescence microscopy analyses, the authors revealed neuronal degeneration and neutrophil infiltration in the ischemic brain parenchyma. Next generation miRNA sequencing identified 19 dysregulated miRNAs known to be linked to stroke and 15 dysregulated miRNAs that were newly detected in the resected tissue from patients. Further miRNA direct target gene assays and bioinformatics analyses demonstrated a strong association of the dysregulated miRNAs with stroke-related biological processes and signaling pathways. While this work is generally well done, validation of target gene expression with RT-qPCR and Western blotting would strengthen the manuscript.

Issues in the manuscript and my suggestions:

-in the abstract, histopathological and immunofluorescence microscopy analyses are mentioned in the Methods but the relevant results are not presented;

-“RT” is used in the text with different meanings. I suggest changing “RT” in lines 129, 131 and 135 to “room temperature”;

-please make “2” in “RT2” superscript (line 170);

-different names are given for the same miR in the text and tables, e.g., “miR-204, miR-17, miR-16, and miR-423” in lines 315-316 vs “miR-204-3p, miR-17-5p, miR-16-5p and miR-423-5p” in Table 1. Of note, miR-423 is also termed miR-423-3p but not miR-423-5p. Please name a miRNA consistently;

-the authors state that additional 22 miR-155 target genes are identified by transcriptome analysis (line 343), but 23 genes are listed (Fig. 5A, green). Please correct the mistake; and

-the gene names are illegible in Supplemental Figure 3.

Author Response

We thank the reviewer for careful and thoughtful evaluation of the manuscript. The point-by-point answers and concerns are addressed below:

-in the abstract, histopathological and immunofluorescence microscopy analyses are mentioned in the Methods but the relevant results are not presented;

We fully agree with the reviewer. Since no quantification has been performed, in the Abstract, we replaced the word “analyses” with the word “evaluation”.

-“RT” is used in the text with different meanings. I suggest changing “RT” in lines 129, 131 and 135 to “room temperature”;

RT replaced with “room temperature”

-please make “2” in “RT2” superscript (line 170);

Corrected

-different names are given for the same miR in the text and tables, e.g., “miR-204, miR-17, miR-16, and miR-423” in lines 315-316 vs “miR-204-3p, miR-17-5p, miR-16-5p and miR-423-5p” in Table 1. Of note, miR-423 is also termed miR-423-3p but not miR-423-5p. Please name a miRNA consistently;

Thank you, miRNA names corrected in accordance with Table 1

-the authors state that additional 22 miR-155 target genes are identified by transcriptome analysis (line 343), but 23 genes are listed (Fig. 5A, green). Please correct the mistake

Corrected

-the gene names are illegible in Supplemental Figure 3.

Due to the high number of miRNA targets, we were not able to make the gene names larger and more visible. The Supplemental Figure 3 is an illustration of the magnitude and variability of genes regulated by each of the detected miRNAs and also, of a high number of shared target genes regulated by different miRNAs. As the material is available in electronic form, we hope that the readers will be able to see the target gene names on the zoomed image.

Reviewer 3 Report

In the article ‘Analysis of human stroke brain tissue resected during decompressive craniectomy/stroke-ectomy surgery’ Andrew P. Carlson et al describe the regulate stroke progression and recovery mediated by miRNA.

I think the manuscript's topics are interesting and will reach the interest of Genes's read.

The introduction is well documented, and the methods are correct; however, I find any points poorly described in the M&M, such as:

    - Version miRBase
    - Version R package
    - Log FC and FDR cutoff for differential expressión

On the other hand, in the results, I recommend that the author draw de cutoff FDR&logFC in the volcano plot. It is not clear the points in this figure with the results shown in table 1.

To sum it put, this work is well organized, and it is easy to read, scientifically sound.

Author Response

We thank the reviewer for the valuable comments. The point-by-point answers are provided below.

    - Version miRBase

miRbase v22 version was used; included in the Methods.
    - Version R package

EdgeR Bioconductor package 3.36.version
    - Log FC and FDR cutoff for differential expression

As stated in Statistical analysis, FDR<0.05 and fold change>2 cutoff was used for differential expression.
On the other hand, in the results, I recommend that the author draw de cutoff FDR&logFC in the volcano plot. It is not clear the points in this figure with the results shown in table 1.

Figure 4 and its legend were substantially modified. Specifically, new Volcano and PCA plots were constructed based on the recommendations from Reviewers 1 and 3.